
# Modelling the High Mercury Wet deposition in the Southeastern US by WRF-GC

Xiaotian Xu[1, a], Xu Feng[2, b], Haipeng Lin[3], Peng Zhang[1], Shaojian Huang[1], Zhengcheng Song[1], Yiming Peng[1], Tzung-May Fu[4], Yanxu Zhang*[1]

[1] Department of Atmospheric Sciences, Nanjing University, Nanjing, Jiangsu, China
[2] Department of Atmospheric and Oceanic Sciences, School of Physics, Peking University, Beijing, China
[3] John A. Paulson School of Engineering and Applied Sciences, Harvard University, Cambridge, Massachusetts, USA
[4] School of Environmental Science and Engineering, Southern University of Science and Technology, Shenzhen, Guangdong, China
[a] Current affiliation: Department of Atmospheric Science, University of Illinois at Urbana-Champaign, Urbana, IL, USA
[b] Current affiliation: John A. Paulson School of Engineering and Applied Sciences, Harvard University, Cambridge, Massachusetts, USA

**Correspondence**: Yanxu Zhang (zhangyx@nju.edu.cn)

**Abstract.** High mercury wet deposition in southeastern United States has been noticed for many years. Previous studies came up with a theory that it was associated with high-altitude divalent mercury scavenged by convective precipitation. Given the coarse resolution of previous models (e.g. GEOS-Chem), this theory is still not fully tested. Here we employed a newly developed WRF-GEOS-Chem (WRF-GC v1.0) model implemented with mercury simulation (WRF-GC-Hg v1.0). We conduct extensive model benchmarking by comparing WRF-GC with different resolutions (from 50 km to 25 km) to GEOS-Chem output ($4° \times 5°$) and data from Mercury Deposition Network (MDN) in July-September 2013. The comparison of mercury wet deposition from two models both present high mercury wet deposition in southeastern United States. We divided simulation results by heights, different types of precipitation and combination of these two variations together and find most of mercury wet deposition concentrates on higher space and caused by convective precipitation. Therefore, we conclude that it is the deep convection caused enhanced mercury wet deposition in the southeastern United States.

## 1 Introduction

Mercury (Hg) is one of the most toxic heavy metals in our environment and undergoes long-range transport (Ariya et al., 2015). It undergoes three major forms in the atmosphere: gaseous elemental mercury (GEM), gaseous oxidized mercury (GOM) and particle-bound mercury (PBM). GEM has extremely low water solubility with a relatively long (~0.5-1 year) residence time in the atmosphere. GEM is slowly oxidized to GOM in the atmosphere initialized by bromine atoms (Holmes et al., 2010), especially in the high-altitudes due to low temperature (Lyman and Jaffe, 2012). While GOM has a much



shorter atmospheric lifetime than GEM due to its strong water solubility and subsequent removal by precipitation (Gonzalez-Raymat et al., 2017; Kaulfus et al., 2017), PBM has similar residence time with GOM due to dry and wet deposition near the source regions (Sexauer Gustin et al., 2012; Coburn et al., 2016).

Wet deposition is a major process for Hg to enter the aquatic and terrestrial ecosystems, wherein causing significant ecological and human health risks (Selin et al., 2007; Rumbold et al., 2019; Fu et al., 2016). The wet deposition flux is thus extensively measured globally, especially in the United States by the Hg Deposition Network (MDN), which was started in 1996 and expanded to contain 81active sites and 117 inactive sites at present-day (Prestbo and Gay, 2009). Previous studies have reported spatial and temporal variation of wet deposition of Hg from over 100 sites spanning from1996 to 2005 and found that Hg wet deposition was high in summer and low in winter and have a distribution that Southeast > Ohio River > Midwest > Northeast. The continuous high-level concentration together with large amount of precipitation every year result in high Hg wet deposition in southeastern region, especially from the Gulf of Mexico to Florida. This level of Hg wet deposition can extent northward to Mississippi Valley. The Hg wet deposition in Midwestern region was relatively moderate and was lowest in northeastern because the precipitation was lower in these areas. Other studies also found out that the Hg wet deposition flux had strong seasonality with a maximum in summer, which was especially true for Florida with approximately 80% of rainfall and Hg wet deposition happening during it (Mason et al., 2000; Fulkerson and Nnadi, 2006; Kaulfus et al., 2017).

One unique phenomenon observed by the MDN sites is the maximum deposition flux over the southeast US, contradicting with that of $NO_3^-$ and $SO_4^{2-}$ that are the maximum over northeast US (http://http://nadp.slh.wisc.edu/data/annualmaps.aspx). The high deposition over this region is hypothesized to be caused by the scavenging of high-concentration GOM in the free troposphere by convective precipitation (Guentzel et al., 2001; Selin et al., 2008). This hypothesis is partially confirmed by Holmes et al., 2016, which found the rain Hg concentrations in seven sites are increased by 50% by thunderstorms relative to weak convective or stratiform events of equal precipitation depth. Kaulfus et al., 2017 found similar patterns for more MDN sites operated in 2005-2013. However, numerical models have trouble reproducing this unique spatial pattern (Holmes et al., 2010), since the global model is generally too coarse to capture deep convective cells that has much smaller spatial scales (Brisson et al., 2016). Later, Zhang et al., 2016a developed a nested-grid simulation of Hg over North America with a higher resolution (1/2° latitude × 2/3° longitude), which improves the model results but still with a significant low bias in this region, leaving an unclosed budget. Here we develop a new Hg simulation capacity with higher resolution based on the WRF-GC model (Feng et al., 2021; Lin et al., 2020). We will further test if the higher (deep) convective precipitation over the southeast US can fully explain the elevated Hg wet deposition fluxes in this region.





## 2 Materials and methods

### 1.1 WRF-GC model with Hg

We develop a new simulation capacity (WRF-GC-Hg v1.0) for atmospheric Hg emission, transport, chemistry, and
deposition based on the WRF-GC v1.0, which is fully described by Lin et al., 2020 and Feng et al., 2021 (For short, keep
WRF-GC for WRF-GC-Hg in the following paragraphs). Briefly, the model contains three parts: the Weather Research
Forecasting (WRF) mesoscale meteorological model (https://www.mmm.ucar.edu/weather-research-and-forecasting-model),
the GEOS-Chem global 3-D atmospheric chemistry model (http://acmg.seas.harvard.edu/geos/) and the WRF-GC coupler.
The WRF v3.9.1.1 (https://github.com/wrf-model/WRF/tree/V3.9.1.1)-Advanced Research WRF (ARW) solver is used to
simulate meteorological processes and the advection of the compositions of the atmosphere with GEOS-Chem v12.2.1
(https://doi.org/10.5281/zenodo.2580198) as a self-contained chemical module. The WRF-GC coupler consists of an
interface, state conversion and management module for the two parent models. In one hand, the WRF-GC model can take
advantage of the WRF model to simulate meteorology in highly customized model domain and resolutions. In addition, the
WRF offers options for configuration, vertical levels, horizontal grids, and map projections. The WRF also supplies options
for land surface physics, planetary boundary layer physics, radiative transfer, cloud microphysics, and cumulus
parameterization (Skamarock et al., 2008). On the other hand, the WRF-GC inherits the state-of-the-art emission, chemistry
and deposition simulation from the GEOS-Chem model.(Eastham et al., 2018; Long et al., 2015) All chemical configurations,
including chemical species, mechanisms, emissions, and diagnostics can be customized using *FlexChem* pre-processor, a
wrapper for the Kinetic-Pre-Processor (KPP) that allows users to add chemical species and reactions and develop their
chemical mechanism (Damian et al., 2002; Sandu and Sander, 2006). The standard chemistry option of GEOS-Chem
includes a full Ox-NOx-VOC-halogen-aerosol chemistry mechanism for the troposphere that contains 208 chemical species
and 981 reactions and a unified tropospheric-stratospheric chemistry extension (UCX) (Eastham et al., 2014).

We add Hg simulation capacity to the WRF-GC model by first introducing Hg species in the GEOS-Chem module: $Hg^0$
(GEM), $Hg^2$ (GOM), HgP (PBM), and two Hg(I) species (HgBr and HgCl). The chemical reactions of Hg that involves the
two-stage oxidation of $Hg^0$ to Hg(I) and $Hg^2$ by halogens, and the reaction rates are following Horowitz et al., 2017a.
Similarly, the aqueous phase reduction of GOM to $Hg^0$ in cloud droplets, and the partitioning of $Hg^2$ and HgP on aerosols are
also included. These Hg species and reactions are added to the standard GEOS-Chem KPP solver so the concentrations of
chemicals that can react with Hg (e.g., Br, BrO, OH, $NO_2$) can be directly read online. Similar to other species in GEOS-
Chem, the emissions of Hg are handled by the Harmonized Emission Component (HEMCO) (Lin et al., 2021). We use the
WHET emission inventory for the anthropogenic emissions of Hg (Zhang et al., 2016b; Horowitz et al., 2017b). The re-
emissions from soil, snow, and ocean are not dynamically modeled but directly read in as a static monthly emission
inventory through HEMCO based on a former GEOS-Chem Hg simulation (Horowitz et al., 2017a).



The WRF-GC model is a regional model that requires initial and lateral boundary conditions, which are provided by a global GEOS-Chem simulation with a consistent setup. In this study, we run the GEOS-Chem Hg simulation at $4° \times 5°$ resolution,

driven by GEOS_FP offline meteorological dataset from the Goddard Earth Observation System (GEOS) of the NASA Global Modeling and Assimilation Office (GMAO) with 47 vertical layers. The GEOS-Chem simulation is configured to start to run a few days earlier than the WRF-GC simulation. The lateral boundary conditions of other species (e.g., Br and $NO_2$) are also provided by a standard GEOS-Chem full chemistry simulation that is driven by the same resolution and meteorological data as the Hg simulation. The output of the GEOS-Chem Hg and full chemistry simulations are then

processed and combined before feeding to the WRF-GC model.

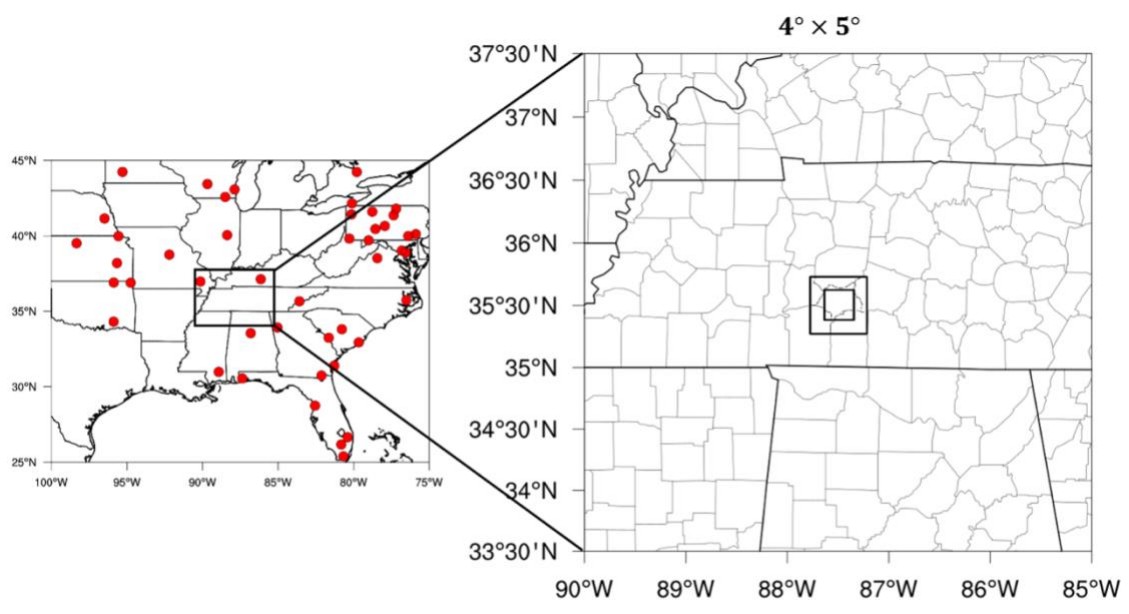

**Fig. 1** Model simulation domain (Left panel: black box represents a single grid of GEOS-Chem $4° \times 5°$ simulation and red dots represents MDN sites within this domain; Right panel: boxes from outside to inside respectively represents one grid of the resolution of $4° \times 5°$, 50km × 50km, 25km × 25km)

We set up a simulation domain over the southeastern US and a simulation period of July-September 2013 because convective precipitation is normally concentrated in summer (Holmes et al., 2016; Fulkerson and Nnadi, 2006). The model domain extends west-east from the middle of Texas to Pennsylvania and north-south from the Canadian border to Florida (Fig. 1). The model horizontal resolution is set as ranged from 50 km to 25 km, with 50 vertical layers. This results in 106 × 111 grid boxes for a horizontal resolution of 25 km, and 51 × 65 boxes for a resolution of 50 km. Table 1 lists the physical

setup and configuration for the WRF model following Feng et al., 2021 and Lin et al., 2020. Large-scale meteorological datasets used for WRF-GC is from National Centers for Environmental Prediction (NCEP) FNL Operational Global



Analysis data at 1°× 1° resolution with 6-hour interval ([doi:10.5065/D6M043C6](doi:10.5065/D6M043C6)). The meteorological data and tracer advection are handled by the WRF model component, while emission, convective transport, chemistry, deposition, and boundary layer mixing are calculated by the GEOS-Chem module. These two model components exchange data online during runtime. This enables WRF-GC Hg simulation to be run at a customized high resolution that stand-alone GEOS-Chem cannot realize. We archive hourly meteorological variables, chemical tracer concentrations, and wet deposition fluxes of $Hg^2$ for analysis.

**Table 1** Physical parameters

| Physics | |
|---|---|
| Microphysics | Morrison Double-Monment scheme(Morrison et al., 2009) |
| Cumulus | New-Tiedtke scheme(M.Tiedtke, 1989) |
| Radiation | RRTMG (both lw & sw)(Iacono et al., 2008) |
| Land Surface | Noah Land Surface Model(Chen and Dudhia, 2001a, b) |
| PBL | Mellor-Yamada Nakanishi Niino scheme(Nakanishi and Niino, 2006) |
| Surface | MM5 Monin-Obukhov(Jiménez et al., 2012) |

Fig. 2 compares the precipitation during July-September 2013 between WRF-GC at different resolutions and CPC Merged Analysis of Precipitation (CMAP) data ($2.5° \times 2.5°$)(Xie and Arkin, 1997). The average total precipitation of WRF-GC 25km × 25km is 3.49 mm/day for the whole simulation region during 3 months in 2013, consistent with the CMAP data (3.16 mm/day). The spatial distribution of the WRF-GC model resembles that of the CMAP data, with the highest precipitation in the northern Gulf of Mexico and extending to the nearby continental regions. The average precipitation over the southeast most region (75°W ~ 95°W, 25°N ~ 35°N) is substantially higher (4.63mm/day), which also agrees with the CMAP data (4.51 mm/day). We further divide the total precipitation from WRF-GC simulation to non-convective (or stratiform) and convective parts. The WRF-GC model suggests that convective precipitation accounts for ~90% of total precipitation in this region (Fig. 2). GEOS_FP offline meteorological dataset also suggests a similar spatial pattern for convective rain compared to CMAP and WRF-GC precipitation, but with much smaller magnitude: accounting for only 35% of total precipitation. A point that cannot be neglected for explaining this problem is that GEOS_FP offline meteorological dataset is the total precipitation of it does not fit in with CMAP observation data well, which needs to be studied further.



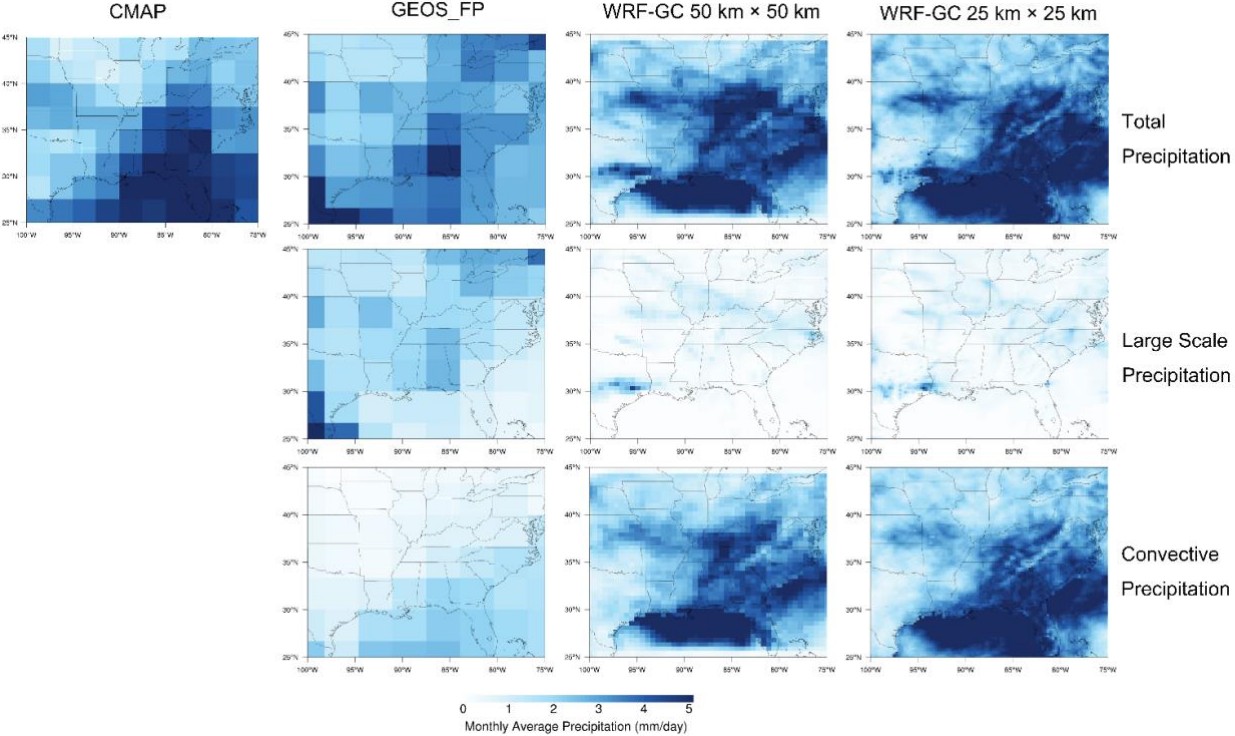

**Fig. 2** Monthly average precipitation from July to September 2013 (Left top corner: CPC Merged Analysis of Precipitation; From second to fourth column: GEOS_FP offline meteorological dataset, WRF-GC precipitation in 50km × 50km, 25km × 25km resolution; From top to bottom: three-month average precipitation, non-convective precipitation, convective precipitation)

The average total precipitation of WRF-GC 25km × 25km is 3.49 mm/day for the whole simulation region, of which convective precipitation and non-convective precipitation accounts for 3.11 mm/day and 0.39 mm/day. However, when the simulation narrows down to the southeastern region (75°W ~ 95°W, 25°N ~ 35°N), the average total precipitation increases to 4.63mm/day and 4.33 mm/day, respectively, while the large-scale precipitation decreases to 0.29 mm/day. This shows that although the southeastern region only takes up 1/3 of whole simulation area, the total precipitation and convective precipitation is 32.66% and 39.23% higher than average, while non-convective are 25.64% lower than the average of whole simulation domain.

### 1.2 Observation data

The weekly-based Hg wet deposition data over the MDN sites are extracted from National Atmospheric Deposition Program (NADP) website (http://nadp.slh.wisc.edu/datalib/mdn/weekly/). The development of MDN has been introduced in introduction part. During the period of this simulation, from July to September 2013, there are over eighty sites inside this





domain having data. Besides, many missing value or unqualified value existed in MDN dataset since it was collected manually. For example, the NE25 site has only three valid data points in three months. Hence it is important to conduct

quality check before using the data. We only take sites that contains at least 70% availability of data for three months. After this quality check, only 55 sites are finally chosen for this study. The atmospheric $Hg^0$ data are extracted from Atmospheric Mercury Network (AMNet) by NADP (http://nadp.slh.wisc.edu/data/AMNet/), and eight AMNet sites are chosen.

## 3 Results and discussion

### 3.1 Comparison of Mercury Concentration between WRF-GC, GEOS-Chem and AMNet

We compare the WRF-GC modelled $Hg^0$ concentrations to AMNet observations to evaluate the model performance (Fig. 3). Due to the relatively long residence time of $Hg^0$, the concentration distributions are relatively uniform in the model domain. The average $Hg^0$ concentrations are 1.25±0.22 ng m$^{-3}$ for the eight sites in the southeast US, while both GEOS-Chem (1.27±0.06 ng m$^{-3}$) and WRF-GC (1.55±0.20 ng m$^{-3}$) models agree with the observations relatively well. The WRF-GC model simulates more elevated $Hg^0$ concentrations in the Ohio River Valley regions than the GEOS-Chem, by which the

coarse resolution smoothens out the higher anthropogenic emissions from mainly utility coal burning (Zhang et al., 2012). Similar patterns are simulated for $Hg^2$ and HgP by WRF-GC due to their shorter residence time in the atmosphere. The influence of large point sources on nearby regions is even more distinct in WRF-GC simulations with higher resolutions. Whereas the GEOS-Chem model cannot capture the hotspots of $Hg^2$ and HgP concentrations associated with point sources largely limited by its resolution. However, both models show substantially higher near-surface $Hg^2$ (GEOS-Chem: 5.98±1.94

165    pg m$^{-3}$, WRF-GC: 14.99±10.99 pg m$^{-3}$, vs AMNet: 3.56±6.09 pg m$^{-3}$). HgP of WRF-GC (4.21±4.13 pg m$^{-3}$) is similar to AMNet: 3.48±2.02 pg m$^{-3}$ and largely higher than GEOS-Chem (0.57±0.42 ng m$^{-3}$). This is likely caused by the potential low sampling bias of the annular denuder coating with potassium chloride (KCl) method(McClure et al., 2014; Lyman et al., 2010; Gustin et al., 2015b) used by AMNet $Hg^2$/HgP measurements, Zhang et al., 2012 compared to the concurrent side-by-side cation exchange membrane measurements (Lyman et al., 2020). Another possible reason is various sampling efficiency

under condition of higher atmospheric ambient ozone and high-level relative humidity caused uncertainties for GOM (Gustin et al., 2013, 2015a; Weiss-Penzias et al., 2015; Huang and Gustin, 2015).

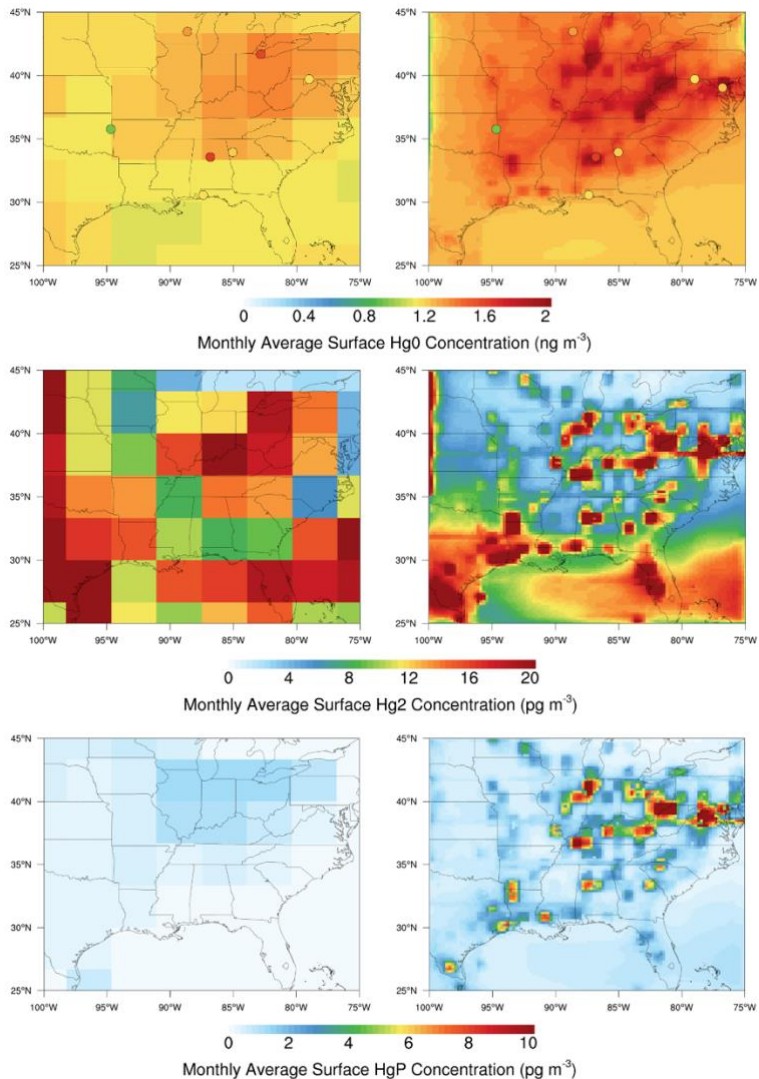

**Fig. 3** Monthly average surface Hg concentration of $Hg^0$ (top), $Hg^2$ (middle), HgP (bottom) from July to September 2013. Dots on the top row of panel represents $Hg^0$ observation data from AMNet of NADP (http://nadp.slh.wisc.edu/AMNet/).

## 3.2 Comparison of Hg Wet Deposition between WRF-GC, GEOS-Chem and MDN

Fig. 4 shows the modelled Hg wet deposition fluxes in the southeast US during July-September 2013, comparing to MDN observations. We include the $Hg^2$ and HgP wet deposition caused by both large-scale (LS or non-convective) and convective (CONV) precipitations. The GEOS-Chem model is included as a benchmark while the WRF-GC at different spatial resolutions (from 50 km to 25 km) are also shown. The MDN sites observed an average of 3.27±1.90 μg m$^{-2}$ for all the 55 sites of the domain in the three-month period. There is a clear spatial pattern for the flux with higher deposition (6.25±1.48





µg m$^{-2}$) over the 12 sites in southeast-most of the US (in the Georgia, Alabama, Mississippi, South Carolina, and Florida states) than the other 43 sites (2.44±0.93 µg m$^{-2}$). Both the GEOS-Chem and WRF-GC simulate similar Hg wet deposition pattern with the observations: 0 to 3 µg m$^{-2}$ in top-left part of the simulation domain, > 4 µg m$^{-2}$ in areas close to the Gulf of Mexico area. However, we find a significant underestimation for these 12 sites by the GEOS-Chem model (3.33 µg m$^{-2}$, 46% lower than MDN). With higher resolutions, the modelled values increase to 2.86±1.07 µg m$^{-2}$ (50 km), 4.16±1.21 µg m$^{-2}$ (25 km), which gradually fix the underestimation as the resolution increase.

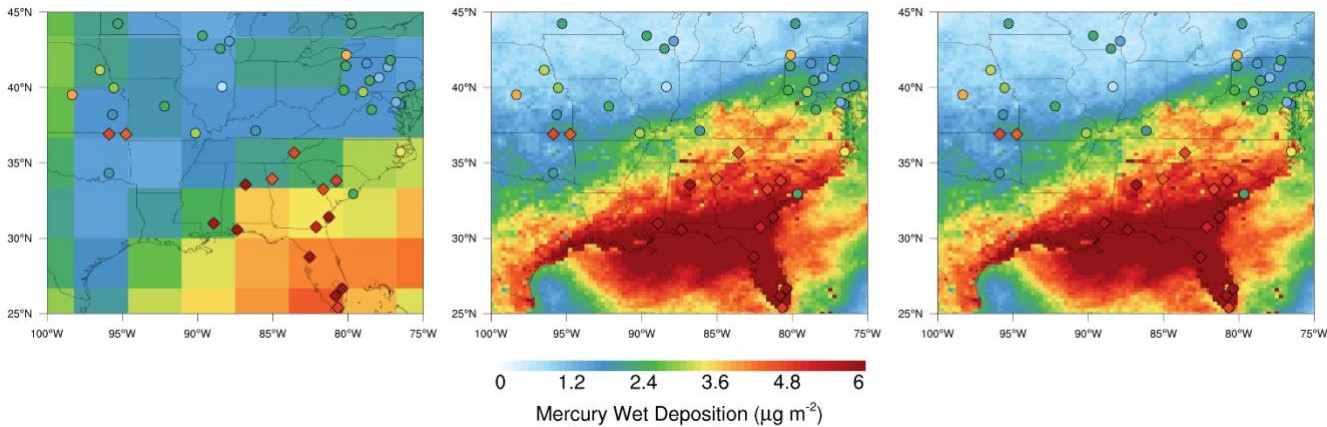

**Fig. 4** Comparison of total Hg Wet Deposition by different model simulation from July to September 2013. The left panel is GEOS-Chem 4° × 5° simulation. Other two column from left to right correspond to different WFF-GC resolution: 50 km × 50 km, 25 km × 25 km. The dots in circle represents wet deposition lower than 4 µg m$^{-2}$ and dots in rhombus represent higher than 4 µg m$^{-2}$.

**3.3 Week-to-week Comparison of Hg Wet Deposition between WRF-GC, GEOS-Chem and MDN**

The MDN sites collect weekly precipitation samples and an ideally total of ~12 samples are included in the three-month period we studied. Fig. 5 compares the measured weekly Hg wet deposition flux over the 12 sites with higher values with the GEOS-Chem and WRF-GC models with different resolutions (plots for the other sites are shown in Fig. S4). The highest deposition fluxes. For example, the FL05 site at Florida state has a total deposition flux of 9.19 µg m$^{-2}$ in the three months, while the largest three weeks (6-27/Aug) contribute 57% with the other 9 weeks contributing only 43%. Similar patterns are also observed in FL34, FL11, MS22, GA40 and SC19.



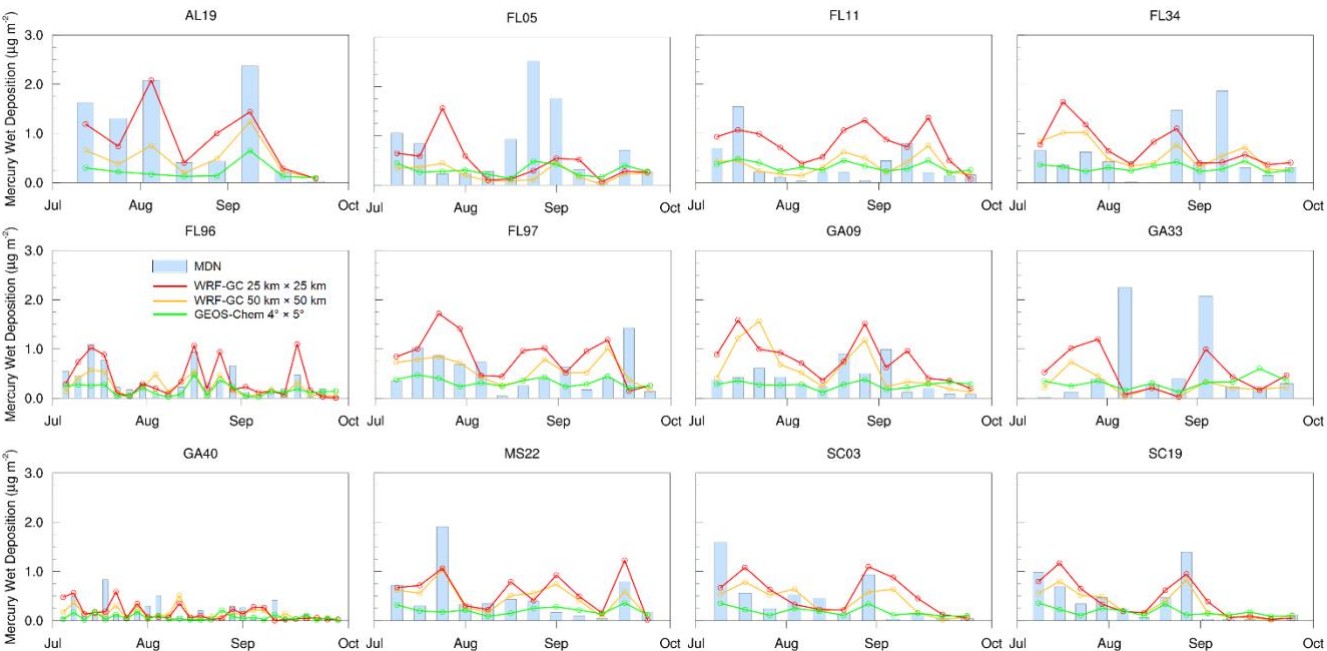

**Fig. 5** Time series plot of comparison of MDN observation, GEOS-Chem 4° × 5° and WRF-GC 50 km × 50 km, 25 km × 25 km simulation results. This plot only shows MDN sites in Florida, full time series plots in SI.

Therefore, we assume that the reason for underestimation of Hg wet deposition in GEOS-Chem is the loss of peak value. For example, the second sampling period of FL11 in Fig. 5, where MDN captures 1.54 µg m$^{-2}$, both GEOS-Chem 4° × 5° and WRF-GC 50 km × 50 km simulated a value of 0.48 µg m$^{-2}$, while WRF-GC 25 km × 25 km shows a value of 0.98 µg m$^{-2}$. As the resolution increases, WRF-GC can better grasp the convective precipitation on a small scale than the GEOS-Chem simulation. However, we find that this increase of resolution is finite. Fig. 6 shows analysis of four short-period cases for 12 high-value MDN sites in July (week 1: 2-9; week 2: 10-16; week 3: 17-23; week 4: 24-30). From GEOS-Chem 4° × 5° to WRF-GC 50 km × 50 km (~0.5°), though GEOS-Chem has better correlation coefficient for most of time, the slope of high-resolution simulation of WRF-GC is much closer to 1:1 line than GEOS-Chem simulation. This result also proves the underestimation of GEOS-Chem simulation in Hg wet deposition. As the WRF-GC resolution increases to 25 km × 25 km (~0.25°), the results are higher than the resolution of 50 km × 50 km, this might be caused by the limitation of resolution of emission inventory (1° × 1°) and even FNL meteorological data (1° × 1°), the difference of resolution between simulation grid and input data may cause large uncertainty.



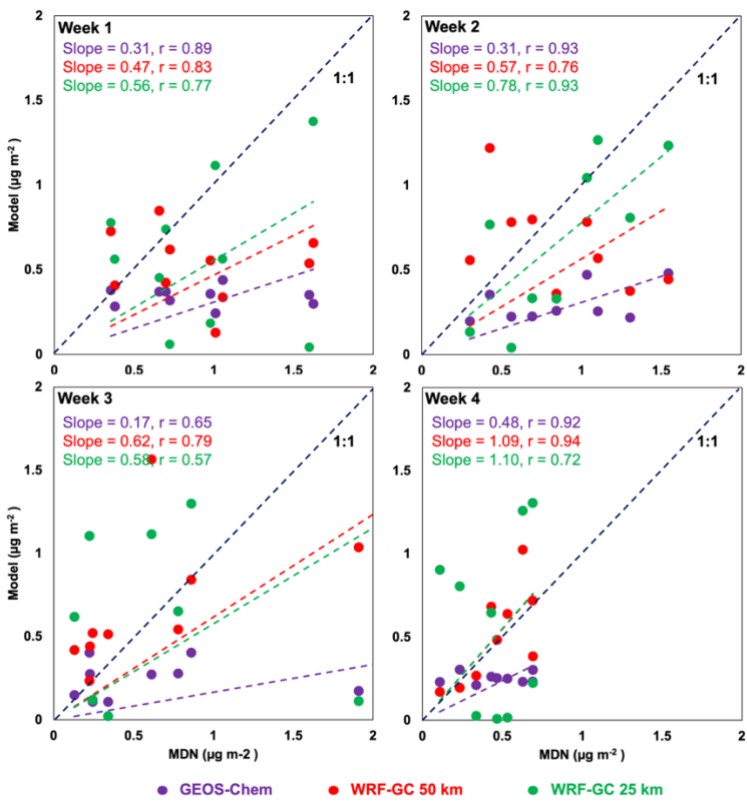

**Fig. 6** Comparison of correlation analysis of different simulation for four separate weeks in 12 high-value MDN sites.

### 3.4 Comparison of Vertical Structure of Hg Wet Deposition between WRF-GC and GEOS-Chem

Fig. 7 shows the vertical structure of total Hg wet deposition simulated by the GEOS-Chem and WRF-GC models. Both GEOS-Chem and WRF-GC presents a rising (4km) trend first and falling (8km), with the highest occurring at ~6km. Hg wet deposition only exists in the border of the Gulf of Mexico and Florida and each model shows Florida has highest value (GEOS-Chem; 0.2 µg m$^{-2}$, WRF-GC: 0.4 µg m$^{-2}$) at this level. At the height increase to ~6 km, the distribution of Hg wet deposition becomes larger with the value of ~0.4 µg m$^{-2}$ for two models and more places have Hg wet deposition larger than 0.4 µg m$^{-2}$. When the height increases to ~8 km, Hg wet deposition in other regions starts to fall, and only southeast-most areas still presents higher value. Although GEOS-Chem 4° × 5° simulation has some difference with WRF-GC 50 km × 50 km and 25 km × 25 km simulation, the whole trend and distribution are similar. Therefore, to better understand which specific type of precipitation caused high Hg wet deposition, we divided the total Hg wet deposition into two types: large-scale-caused Hg wet deposition (LS or non-convective) and convective-caused Hg wet deposition (CONV).

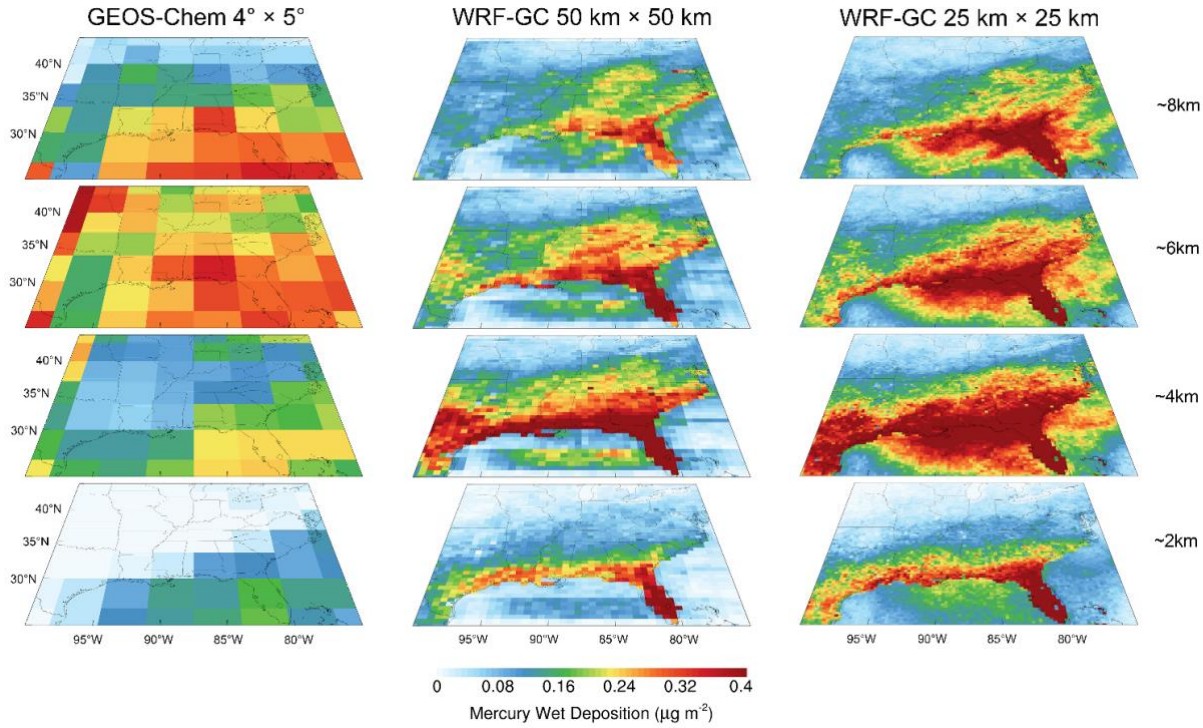

**Fig. 7** Comparation of total Hg wet deposition of GEOS-Chem and WRF-GC at different level and resolution. From top to bottom is the simulation results for ~2 km, ~4 km, ~6 km, ~8 km level, respectively. The first column is GEOS-Chem 4° × 5° simulation results. Other two from left to right correspond to different WFF-GC resolution: 50 km × 50 km, 25 km × 25 km.

### 3.5 Comparison of Different Type of Hg Wet Deposition between WRF-GC and GEOS-Chem

The first and second row of Fig. 8 show Hg wet deposition caused by LS and CONV, respectively. LS of GEOS-Chem is slightly higher than that of WRF-GC, but we can still clearly see the higher value of > 2 µg m$^{-2}$ distributed in the southeast-most area. However, for CONV, although two models share higher Hg wet deposition in the same area, but CONV of GEOS-Chem is lower than 1.8 µg m$^{-2}$, where CONV of WRF-GC is normally higher than 3 µg m$^{-2}$. Besides, we calculated the percentage of LS, CONV, and ratio of LS/CONV from different model, respectively. CONV in GEOS-Chem only take 23.41% of total Hg we deposition in this domain, while WRF-GC has 61.54% of Hg wet deposition resulting from CONV. The ratio of LS/CONV of GEOS-Chem is 3.27 and of WRF-GC is 0.56. This both preliminarily verified that Hg wet deposition in southeastern US came from convective precipitation. To further prove the height of convective precipitation that caused high Hg wet deposition, we divided these two types of Hg wet deposition by height.

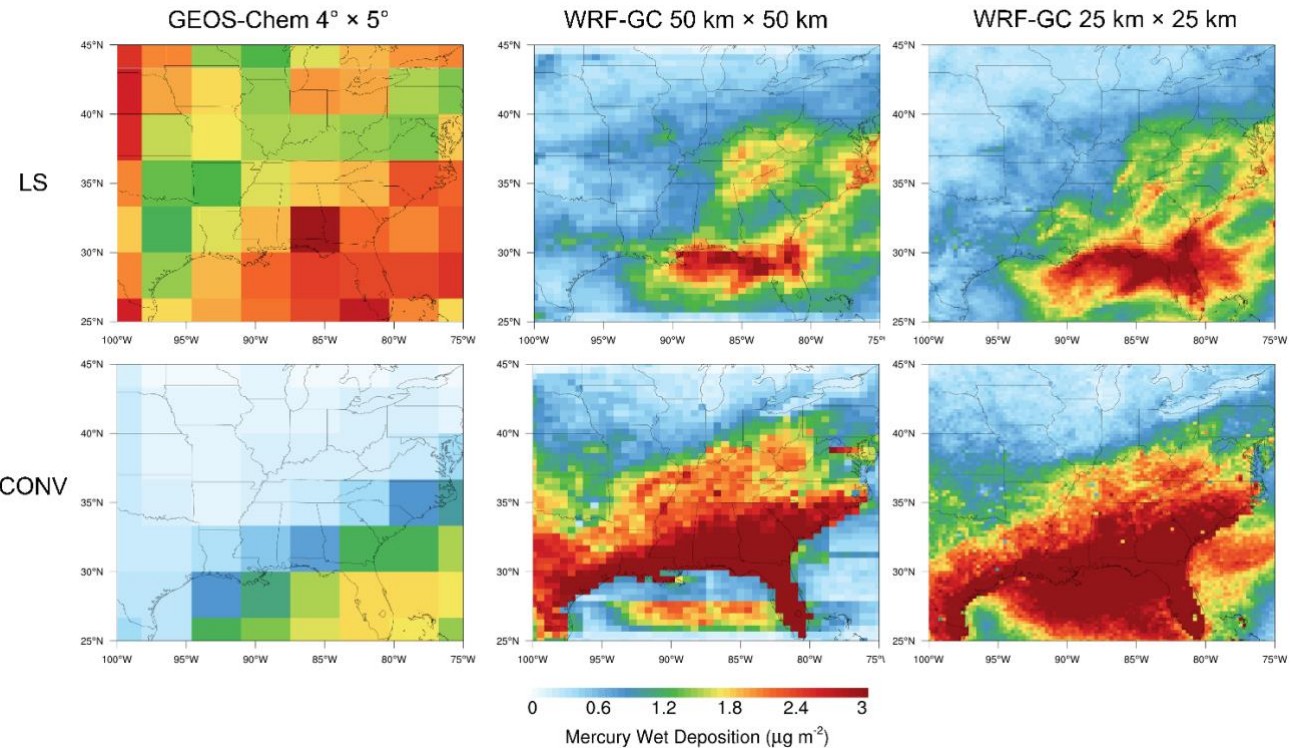

**Fig. 8** Comparison of different type of wet deposition of GEOS-Chem and WRF-GC. The top panel shows LS, and the bottom panel shows CONV. The first column is GEOS-Chem $4° × 5°$ simulation results. Other column from left to right

correspond to different WFF-GC resolution: 50 km × 50 km, 25 km × 25 km.

### 3.6 Comparison of Vertical Structure for Different Type of Hg Wet Deposition between WRF-GC and GEOS-Chem

Fig. 9 shows Hg wet deposition by LS from GEOS-Chem and WRF-GC at different resolution and height. LS from GEOS-Chem and WRF-GC both increase as the height increase and the two models all have value < 0.1µg m$^{-2}$ under ~6km. However, LS from GEOS-Chem is much larger than WRF-GC at height of ~6 km, we assume this might be cause by the

250 GEOS_FP meteorological data because large scale precipitation is stronger than WRF-GC in Fig. 2. LS at ~8 km is basically same for two model, but as the resolution increase, the description of distribution of Hg wet position is getting better. Fig. 10 shows Hg wet deposition by CONV from GEOS-Chem and WRF-GC at different resolution and height. We can see the higher CONV of two model both distributed in southeast-most area and presents an increasing trend until ~4 km and decrease later. CONV of GEOS-Chem is all lower than 0.15 µg m$^{-2}$ whilst WRF-GC can reach to 0.8 µg m$^{-2}$ at ~4 km, 0.5 µg

m$^{-2}$ at ~6 km and remain 0.3 µg m$^{-2}$ at ~8 km. Besides, by comparing the different resolution of WRF-GC simulation, the distribution of Hg wet deposition is getting more and more continuous. Also, because higher resolution can capture the peak





Hg wet deposition by convective precipitation in a small domain, the total Hg wet deposition slightly increase with the resolution.

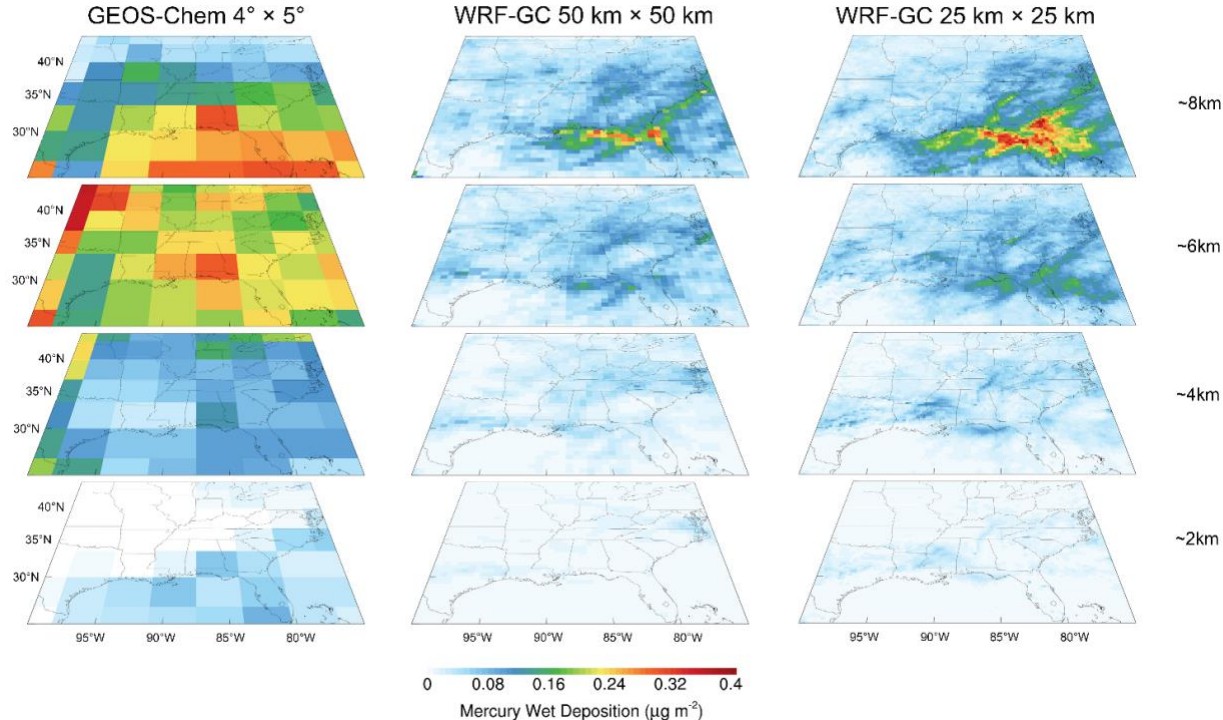

**Fig. 9** Comparison of LS of GEOS-Chem and WRF-GC at different levels and resolutions. From top to bottom stands for Hg wet deposition at ~2 km, ~4 km, ~6 km, ~8 km, respectively. The first column is GEOS-Chem $4° \times 5°$ simulation results. Other columns from left to right correspond to different WFF-GC resolution: 50 km × 50 km, 25 km × 25 km.

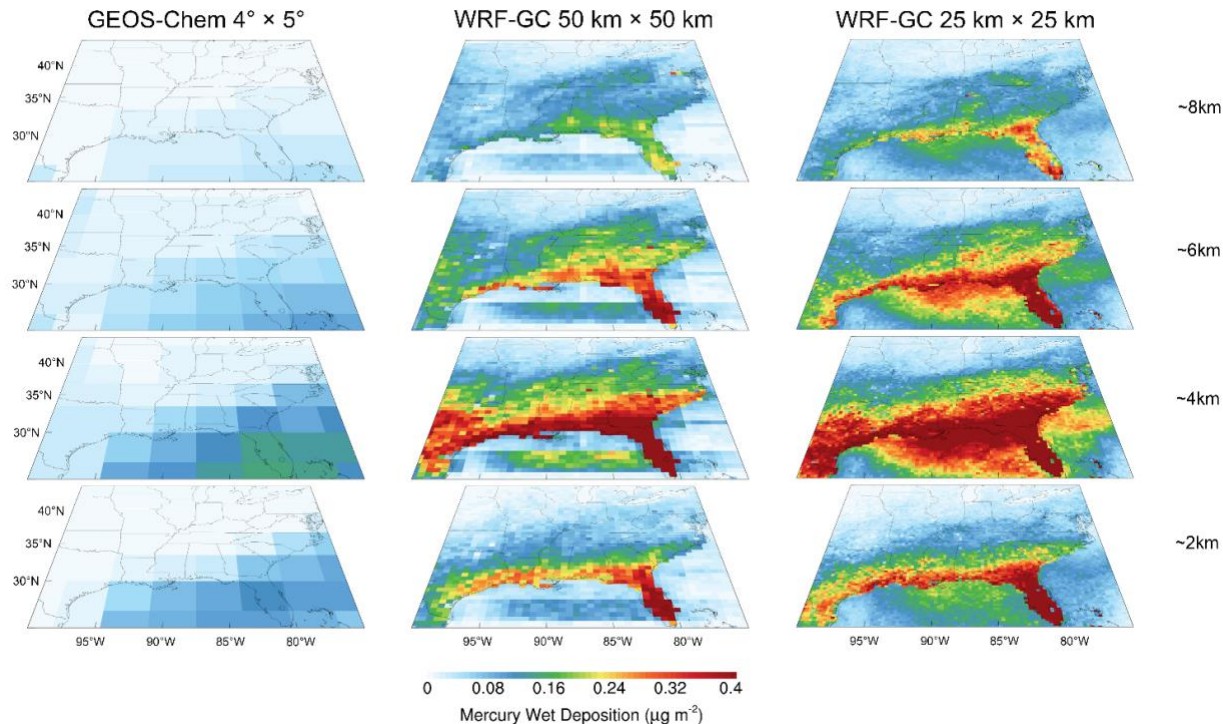

**Fig. 10** Comparison of CONV of GEOS-Chem and WRF-GC at different levels and resolutions. From top to bottom
stands for Hg wet deposition at ~2 km, ~4 km, ~6 km, ~8 km, respectively. The first column is GEOS-Chem $4^\circ \times 5^\circ$
simulation results. Other columns from left to right correspond to different WFF-GC resolution: 50 km × 50 km, 25 km × 25
km.

**Conclusion**

This study applies a new coupled WRF-GC v1.0 model and develops comprehensive codes of Hg simulation into the model
(WRF-GC-Hg v1.0) to explain the reason of higher wet deposition in southeastern United States. Boundary conditions are
provided by a global GEOS-Chem Hg simulation at $4^\circ \times 5^\circ$ resolution with same emissions and chemistry.

Comparisons between WRF-GC simulation in 50km × 50km, 25km × 25km resolution, GEOS-Chem Hg simulation results
at $4^\circ \times 5^\circ$ resolution and observation dataset from AMNet and MDN were extensively conducted. WRF-GC simulated an
average $Hg^0$ concentration of 1.61±0.20 ng m$^{-3}$, which agree with GEOS-Chem simulation 1.20±0.06 ng m$^{-3}$ and AMNet
observation 1.25±0.22 ng m$^{-3}$. There is a large difference of $Hg^2$/HgP concentration from AMNet to two models, which we
suggest it is caused by the potential low sampling bias of the traditional annular denuder coating with potassium chloride
method used by AMNet $Hg^2$/HgP measurements.

Regarding Hg wet deposition, two models have similar distribution in southeast-most area, but the value of Hg wet
deposition of WRF-GC (3.48±2.02 µg m$^{-2}$) is closer to MDN sites (3.27±1.90 µg m$^{-2}$) than GEOS-Chem (1.25±0.22 µg m$^{-2}$).



The higher value usually happens in this area, so here we chose 12 sites with higher value in states of Mississippi, Alabama,
Georgia, South Carolina, and Florida. After analysing the time series variation, we found that Hg wet deposition came from
a few short periods but not evenly distributed in three months, which corresponding to occurrence of convective
precipitation.

To prove the higher Hg wet deposition came from convective precipitation at higher space. We first describe Hg wet
deposition with difference model at different height. It is clearly that Hg wet deposition from two model increase with height
first and then decrease, and the most of Hg wet deposition was in higher height. Then we divided Hg wet deposition by
different type of precipitation: large-scale and convective. LS of GEOS-Chem is slightly higher than that of WRF-GC, but
we can still clearly see the higher value of $> 2$ µg m$^{-2}$ in southeast-most area. However, CONV of GEOS-Chem lower than
1.8 µg m$^{-2}$ while that of WRF-GC are normally higher than 3 µg m$^{-2}$. Besides, the ratio of LS/CONV from GEOS-Chem is
3.27 and of WRF-GC is 0.56 since CONV in GEOS-Chem only take 23.41% of total Hg we deposition in this domain, while
WRF-GC has 61.54% of Hg wet deposition. At last, we combine two abovementioned analysis and elaborate Hg wet
deposition by different types of precipitation at different height. LS from GEOS-Chem and WRF-GC both increase as the
height increase and two models all have value $< 0.1$µg m$^{-2}$ under ~6km, whilst LS from GEOS-Chem is much larger than
WRF-GC at height of ~6 km, we assume GEOS_FP meteorological data might cause this situation. CONV from GEOS-
Chem and WRF-GC both distributed in southeast-most area and presents an increase trend until ~4 km and decrease later.
However, CONV of GEOS-Chem is all lower than 0.15 µg m$^{-2}$ whilst WRF-GC can reach to 0.8 µg m$^{-2}$ at ~4 km, 0.5 µg m$^{-2}$
at ~6 km and remain 0.3 µg m$^{-2}$ at ~8 km. This may slightly be different from previous research that high Hg wet deposition
was scavenged by supercell thunderstorm at height of over 10 km.

In addition, by comparing the different resolution of WRF-GC simulation, the distribution of Hg wet deposition is getting
more and more continuous. Also, because higher resolution can grasp the peak Hg wet deposition by convective
precipitation in a small domain, the total Hg wet deposition slightly increase with the resolution. However, we need to notice
that the increase of simulation performance with increase of resolution is finite.

*Code availability*

The parent WRF-GC v1.0 model is open source and can be accessed at http://wrf.geos-chem.org or downloaded from
GitHub (https://github.com/jimmielin/wrf-gc-release/tree/v0.9). The code used for implementing mercury into WRF-GC
(WRF-GC-Hg v1.0) in this paper can be obtained from GitHub (https://github.com/Jim-Xu/WRF-GC-Hg). The WRF-GC-
Hg v1.0 is permanently archived at https://doi.org/10.5281/zenodo.5787321 (last access: 16th December 2021).



*Author contribution*

YZ supervised and guided the whole project, XX did all simulations, analysis, and paper writing, XF, HL & TMF provides many technical advice during the code development, PZ & SH & ZS  and YP provides advices and assistance in analyzing
results.

*Competing interests*

The authors declare no competing interest.





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
