# Peer review of "Modeling the High Mercury Wet deposition in the Southeastern US by WRF-GC-Hg v1.0"

_Geoscientific Model Development, 2021_

## Author Comment (AC2)

gmd-2021-404

We would like to thank the reviewers for the constructive comments. This revised manuscript has been further improved based on the reviewers' comments and suggestions. The point-by-point responses to each reviewer's comments are given below.

**Referee 1**

General Comments

The manuscript describes the development of WRF-GC-Hg model, based on the already developed WRF-GC model and also shows a case study of the application of WRF-GC-Hg model in understanding the high Hg wet deposition in Southeast US. I do have several major comments:

- Does the paper focus on the study of the high Hg wet deposition in Southeast US only or also the development of the WRF-GC-Hg model? I would suggest including the development of the WRF-GC-Hg model as an important component of the paper. If so, the title should be revised as something: Development of WRF-GC-Hg v1.0 and its application in studying Hg wet deposition in Southeast US. This will make the paper stronger and more applicable. With that, the paper will need to be reorganized to include one part to focus on the new development and its evaluation and another part to focus on the study of the high Hg wet deposition in the Southeast US. It will be also nice to extend the domain to the whole continental US for the WRF-GC-Hg evaluation part.

- We thank the reviewer for this helpful suggestion. Our motivation is to study high mercury wet deposition in southeastern US. The reason we chose WRF-GC is that this model has many advantages (like flexible resolution, better meteorology simulation, HEMCO inside the model for emission inventories, etc.)

  However, given that WRF-GC is a novel model without complimentary mercury libraries, some developments have been done for this research, so here we added some paragraphs to better describe the WRF-GC-Hg v1.0.

  Corresponding to our paper, the modifications are as follows:
  1. The advantage of WRF-GC.
     Line 64-70: Therefore, we chose WRF-GC (Feng et al., 2021; Lin et al., 2020) to develop a new Hg simulation capacity with a complimentary Hg library because WRF-GC has several advantages: 1) It has flexible resolution and widely accepted meteorology simulation provided by WRF model; 2) The Hg chemistry included by GEOS-Chem model is more up-to-date than many other models (Horowitz et al., 2017); 3) It is relatively easy to port Hg library from GEOS-Chem to WRF-GC-Hg.

  2. We added a figure to illustrate the structure of WRF-GC-Hg v1.0 and our modifications.
     Line 74: The model's framework is shown in Fig. 1.

**Fig. 1** WRF-GC-Hg v1.0 framework based on WRF-GC v1.0 (Lin et al., 2020)

3. Chemistry library
   Line 92: We implement a complimentary Hg chemistry library (Fig. 1) to the WRF-GC model by first introducing …

4. Emission module
   Line 99-100: We use the WHET emission inventory ($1° \times 1°$) for the anthropogenic Hg emissions (Zhang et al., 2016) as well as natural emissions and re-emissions inventory ($4° \times 5°$) from Horowitz et al. (2017) (see Fig. 1).

- Regarding the domain of simulation, previous research (Holmes et al., 2016; Fulkerson and Nnadi, 2006) have shown the underestimation of Hg wet deposition in the southeastern US. Therefore, we only chose the southeastern US region for this research considering that WRF-GC high-resolution simulation is highly computationally intensive.

- The paper conducts two WRF-GC-Hg sensitivity simulations with a horizontal resolution of 50 km and 25 km respectively and compare the results to the GEOS-Chem simulation with a spatial resolution of 4º x 5º. As expected, the WRF-GC-Hg simulations with a finer spatial resolution will resolve more spatial signals. The comparison will be more meaningful to include the GEOS-Chem nested Hg simulation results, which are comparable to WRF-GC-Hg simulations.
Thanks for your suggestion. We had considered your suggestion before, but we find the currently existing GEOS-Chem Hg nested-grid simulation over North America is out of date (Zhang et al, 2012). The simulation is based on chemistry from Holmes et al. (2010), but we use an updated Hg chemistry from Horowitz et al. (2017). Therefore, the GEOS-Chem simulation nested Hg simulation is not off-the-shelf but would still require significant development and validation works. In addition, our main focus is not to compare GEOS-Chem and WRF-GC, but to find the simulation performance improvement with increasing resolution. We also expect the changes caused by increasing resolution to be similar between different models. We thus think it should

be enough to compare a coarse resolution of GEOS-Chem with different resolutions of WRF-GC-Hg.

- Some of the sentences are confusing. I would suggest improving the English language throughout the manuscript.

Thanks for your suggestion. A grammar check has been done during this revision. For major changes, we have highlighted contents with yellow color.

Specific comments

1. Line 23-26, can you be more explicit here about the heights, different types of precipitation etc.?

Line 23-26: We divided simulation results by heights (2km, 4km, 6km, 8km), different types of precipitation (large-scale and convective), and combinations of these two variations together and find most of mercury wet deposition concentrates on higher level and caused by convective precipitation.

2. Line 28: It is atmospheric Hg that can undergo long-range transport, it is not accurate to say that Hg goes through long-range transport here.

Line 29-30: Mercury (Hg) is one of the most toxic heavy metals in our environment. Atmospheric Hg can undergo long-range transport (Ariya et al., 2015) in three major forms: gaseous elemental mercury (GEM), gaseous oxidized mercury (GOM), and particle-bound mercury (PBM).

3. Line 46-48: I am not sure what it means here. What is the 80% of rainfall?

Line 48: approximately 80% of rainfall amount

4. Line 108: The model is run as one domain with 50 km and 25 km, right? "The model horizontal resolution is set as ranged from 50 km to 25 km, with 50 vertical layers." It sounds like the model is run as a nested domain. Please clarify here.

Line 120: We ran simulations with different horizontal resolutions (50 km × 50 km and 25 km × 25 km for WRF-GC and 4° × 5° for GEOS-Chem) rather than using nested domains.

5. Line 120: what are the CMAP data? Are they merging observation and model data? Please provide more information here.

Line 133-134: The CMAP is 2.5° × 2.5° monthly analyses of global precipitation, generated from merging rain gauges and several satellite-based algorithms (Xie and Arkin, 1997).

6. Line 131-132: I do not quite understand the sentence here.

This is a problem we found during analyzing the data, but it is not related to this research, so we deleted it.

7. In Table 1, please spell out the lw and sw.

Table 1: Longwave and shortwave

8.      Line 140: the average total precipitation increases to 4.63 mm/day and 4.33 mm/day, so is it 4.63 or 4.33?

Apologies for not clarifying this. What we are trying to express is when we narrow down the domain to the southeast-most region, the changes in total precipitation, convective precipitation and large-scale precipitation.

Line 149-151: the average total precipitation increases to 4.63mm/day and convective precipitation increases to 4.33 mm/day, while the large-scale precipitation decreases to 0.29 mm/day.

9.      Line 152: Are the eight AMNet sites shown in Figure 1? If so, can you use different symbols to differentiate them from the MDN sites?

We added AMNet sites with different symbols to Fig. 2.

[Figure]

Fig. 2 Model simulation domain (Left panel: black box represents a single grid of GEOS-Chem $4° \times 5°$ simulation, red circle dots represent MDN sites and triangle dots represents AMNet sites within this domain; Right panel: boxes from outside to inside respectively represents one grid of the resolution of $4° \times 5°$, 50km $\times$ 50km, 25km $\times$ 25km)

10.     Line 205: However, we find that this increase of resolution is finite. What do you mean here?

Line 225-226: However, we find that this increase of resolution is finite because the improvement of the increase of wet deposition flux is not that obvious as WRF-GC resolution increases.

11.     Line 254: whilst?

Here we want to compare CONV between different models at the same time. To better understand this sentence, we replaced "whilst" to "but".

Line 275: CONV of GEOS-Chem is all lower than 0.15 µg m$^{-2}$ but WRF-GC can reach 0.8 µg m$^{-2}$ at ~4 km

12.     Line 156-158, 273-275: WRF-GC-Hg simulated Hg$^0$ concentration is 1.61±0.20 ng m$^{-3}$, which does not quite agree with the GEOS-Chem and observation concentration. Do you know why WRF-GC-Hg simulated higher Hg$^0$ concentration than both GEOS-

Chem and the ground observation, even though WRF-GC-Hg simulated better Hg wet deposition? It is due to the atmospheric redox chemistry or something else?

The development of WRF-GC-Hg (Hg chemistry library) coupling GEOS-Chem full-chemistry library with offline Br simulation. Even though all parameters were set the same as running GEOS-Chem, aqueous reductions and aerosol concentration may not be the same as GEOS-Chem's results.

Line 168-172: The average $Hg^0$ concentrations are $1.25\pm0.22$ ng m$^{-3}$ for the eight sites in the southeast US, which agree well with GEOS-Chem results of $1.27\pm0.06$ ng m$^{-3}$. WRF-GC ($1.61\pm0.20$ ng m$^{-3}$) model does not agree with the observations or GEOS-Chem relatively well but is close. This might be due to the development of WRF-GC (Hg chemistry library) coupling GEOS-Chem full-chemistry library with offline Br simulation. Even though all parameters were set the same as running GEOS-Chem, aqueous reductions and aerosol concentration may not be the same as GEOS-Chem's results.

13.     Line 279: in this area, which area do you mean here?

Line 300-301: Southeast-most area (states of Mississippi, Alabama, Georgia, South Carolina, and Florida)

14.     Line 450-454: (Zhang et al., 2016a) and (Zhang et al., 2016b) are the same.

Sorry for the mistake, we have corrected the reference by deleting the repeated one.

15.     Figure 7, 9, 10 are hard to read.

What we tried to express here is the comparison between different models at different heights (~2,4,6,8 km). From left to right is a comparison between different models. From top to bottom is a comparison between different heights. Here, we want to use this comparison to better see the distribution of Hg wet deposition on different levels. We moved the labels for heights to the left of the figure and added compasses to different panels.

Here are the changes for the three figures:

[Figure]

**Fig. 8** Comparation of total Hg wet deposition of GEOS-Chem and WRF-GC at different levels and resolution. From top to bottom is the simulation results for ~2 km, ~4 km, ~6 km, ~8 km level, respectively. The first column is GEOS-Chem 4° × 5° simulation results. The other two from left to right correspond to different WFF-GC resolutions: 50 km × 50 km, 25 km × 25 km.

[Figure]

**Fig. 10** Comparison of LS of GEOS-Chem and WRF-GC at different levels and resolutions. From top to bottom stands for Hg wet deposition at ~2 km, ~4 km, ~6 km, ~8 km, respectively. The first column is GEOS-Chem 4° × 5° simulation results. Other columns from left to right correspond to different WFF-GC resolutions: 50 km × 50 km, 25 km × 25 km.

[Figure]

**Fig. 11** Comparison of CONV of GEOS-Chem and WRF-GC at different levels and resolutions. From top to bottom stands for Hg wet deposition at ~2 km, ~4 km, ~6 km, ~8 km, respectively. The first column is GEOS-Chem 4° × 5° simulation results. Other columns from left to right correspond to different WFF-GC resolutions: 50 km × 50 km, 25 km × 25 km.

16.     In section 3.2-3.3: did you compare the model simulated precipitation vs precipitation measured at MDN sites?

Thanks for your suggestion. Here, we did not compare model simulated precipitation to MDN sites' measured precipitation because MDN sites only have total precipitation data. What we focus on in this paper is the Hg wet deposition caused by different types of precipitation. Thus, we thought it should not be necessary to compare precipitation site by site.

To prove our assumption, Fig. 3 shows the comparison of precipitation between CMAP, GEOS_FP meteorological dataset, and WRF-GC-Hg simulation results, with different types of precipitation (total, large-scale, and convective precipitation).

[Figure]

Fig. 3 Monthly average precipitation from July to September 2013 (Left top corner: CPC Merged Analysis of Precipitation; From the second to fourth column: GEOS_FP offline meteorological dataset, WRF-GC precipitation in 50km × 50km, 25km × 25km resolution; From top to bottom: three-month average precipitation, non-convective precipitation, convective precipitation)

17.    This work only focuses on only one year (2013) study, what do you think of the interannual variability of the precipitation and Hg wet deposition?
Although WRF-GC has many advantages, as we mentioned before, for this research, when we increase the resolution to 25 km × 25 km, a three-month simulation was already computationally intensive. With this, we only applied the year 2013 as a case study, but it was able to prove our assumption.  In our future work, we definitely could simulate more years.

Reference:
Lin, H., Feng, X., Fu, T.-M., Tian, H., Ma, Y., Zhang, L., Jacob, D. J., Yantosca, R. M., Sulprizio, M. P., Lundgren, E. W., Zhuang, J., Zhang, Q., Lu, X., Zhang, L., Shen, L., Guo, J., Eastham, S. D., and Keller, C. A.: WRF-GC (v1.0): online coupling of WRF (v3.9.1.1) and GEOS-Chem (v12.2.1) for regional atmospheric chemistry modeling – Part 1: Description of the one-way model, 13, 3241–3265, https://doi.org/10.5194/gmd-13-3241-2020, 2020.

Zhang, Y., Jaeglé, L., Van Donkelaar, A., Martin, R. V., Holmes, C. D., Amos, H. M., Wang, Q., Talbot, R., Artz, R., Brooks, S., Luke, W., Holsen, T. M., Felton, D., Miller, E. K., Perry, K. D., Schmeltz, D., Steffen, A., Tordon, R., Weiss-Penzias, P., and Zsolway, R.: Nested-grid simulation of mercury over North America, 12, 6095–6111, https://doi.org/10.5194/acp-12-6095-2012, 2012.

Holmes, C. D., Jacob, D. J., Corbitt, E. S., Mao, J., Yang, X., Talbot, R., and Slemr, F.: Global atmospheric model for mercury including oxidation by bromine atoms, Atmos. Chem. Phys., 10, 12037–12057, doi:10.5194/acp-10-12037-2010, 2010

Horowitz, H. M., Jacob, D. J., Zhang, Y., DIbble, T. S., Slemr, F., Amos, H. M., Schmidt, J. A., Corbitt, E. S., Marais, E. A., and Sunderland, E. M.: A new mechanism for atmospheric mercury redox chemistry: Implications for the global mercury budget, 17, 6353–6371, https://doi.org/10.5194/acp-17-6353-2017, 2017.

Xie, P. and Arkin, P. A.: Global Precipitation: A 17-Year Monthly Analysis Based on Gauge Observations, Satellite Estimates, and Numerical Model Outputs, 78, 2539–2558, https://doi.org/10.1175/1520-0477(1997)078<2539:GPAYMA>2.0.CO;2, 1997.

Ariya, P. A., Amyot, M., Dastoor, A., Deeds, D., Feinberg, A., Kos, G., Poulain, A., Ryjkov, A., Semeniuk, K., Subir, M., and Toyota, K.: Mercury Physicochemical and Biogeochemical Transformation in the Atmosphere and at Atmospheric Interfaces: A Review and Future Directions, 115, 3760–3802, https://doi.org/10.1021/cr500667e, 2015.

Holmes, C. D., Krishnamurthy, N. P., Caffrey, J. M., Landing, W. M., Edgerton, E. S., Knapp, K. R., and Nair, U. S.: Thunderstorms increase mercury wet deposition, 50, 9343–9350, https://doi.org/10.1021/acs.est.6b02586, 2016.

Fulkerson, M. and Nnadi, F. N.: Predicting mercury wet deposition in Florida: A simple approach, 40, 3962–3968, https://doi.org/10.1016/j.atmosenv.2006.02.028, 2006.

---

## Author Comment (AC3)

gmd-2021-404

We would like to thank the reviewers for the constructive comments. This revised manuscript has been further improved based on the reviewers' comments and suggestions. The point-by-point responses to the reviewer's comments are given below.

**Referee 2**

This study presents a representative demonstration of GEOS-CHEM multi-phase, multi-species mercury atmospheric chemical transport algorithms in the WRF-GEOS-Chem model and addresses one extant scientific question through multi-scale regional simulations. This novel tool is a major incremental advance in regional mercury modeling capabilities in scope for GMD and for scientific questions within the scope of EGU. This is also a first demonstration of the ease of transferring GEOS-CHEM algorithms to WRF-GC without the additional model development, porting, and QA/QC efforts usually required, another large incremental advance in regional atmospheric chemical transport modeling beyond the mercury application shown here. Thus, the title reflects the contents of the article as demonstration of model development. The overall presentation well-structured and clearly written, and ready for copy editing for fluency and precision. The original contributions are highlighted, and the number and quality of references are appropriate.

The manuscript nearly meets criteria for initial model demonstration, with a few important omissions. I recommend acceptance following minor revisions to evidence, context, methods and assumptions, and supplemental materials to support reproducibility:

> 1.      Comparison of capabilities and performance to prior regional Hg modeling in the domain and elsewhere is a basic expectation, and notably absent. First, the authors must identify the differences and advantages of this model's atmospheric chemical transport mechanisms to other regional models that resolve Hg (e.g., CMAQ, CAMx, WRF-CHEM, STEM-Hg), cite them, and clearly communicate the value and novelty. A table comparing mechanism features would help. Performance comparison to regional scientific and regulatory modeling over the study domain is essential to communicating the applied value of this tool and would best be achieved by comparing to contemporary community model performance benchmarks (e.g., Emery et al., 2017) for the same season. Qualitative and quantitative summaries of the performance advantages of this regional model should appear in the abstract.

Thanks for your suggestion. We added content to better describe the difference between models and why we chose WRF-GC as our working model:

First, WRF-GC takes advantage of flexible resolution, better meteorology simulation from WRF, and updated Hg chemistry library from GEOS-Chem, while other models have not updated Hg chemistry mechanisms for a long time.

Second, this model will potentially benefit a large number of users that are familiar with WRF-Chem and GEOS-Chem.

Third, the WRF-GC-Hg model has relatively good portability. WRF-GC is off-the-shelf ready to work, and all we need to do is port GEOS-Chem Hg chemistry to WRF-GC-Hg.

Fourth, we wanted to focus on a scientific problem of high Hg wet deposition in the southeastern US, so we did not spend much effort on the intercomparison of simulation performance between different models.

However, considering that WRF-GC-Hg is a novel model for Hg simulation, we decided to add more description of its development in the revised paper:

Line 60-70: Except for GEOS-Chem (Zhang et al., 2012), Hg simulation was implemented into many models like WRF-Chem (Gencarelli et al., 2014), CMAQ (Bullock and Brehme, 2002), and STEM-Hg (Pan et al., 2010), etc. Models like WRF-Chem and CMAQ also use WRF for meteorology simulation with different Hg chemistry libraries that have not been updated in recent years. Therefore, we chose WRF-GC (Feng et al., 2021; Lin et al., 2020) to develop a new Hg simulation capacity with a complimentary Hg library because WRF-GC has several advantages: 1) It has flexible resolution and widely accepted meteorology simulation provided by WRF model; 2) The Hg chemistry included by GEOS-Chem model is more up-to-date than many other models (Horowitz et al., 2017); 3) It is relatively easy to port Hg library from GEOS-Chem to WRF-GC-Hg.

[Figure]

**Fig. 1** WRF-GC-Hg v1.0 framework based on WRF-GC v1.0 (Lin et al., 2020)

Line 76: The model's framework is shown in Fig. 1.

Line 94: We implement a complimentary Hg chemistry library (see Fig. 1) to the WRF-GC model by first introducing …

Line 101-102: We use the WHET emission inventory (1° × 1°) for the anthropogenic Hg emissions (Zhang et al., 2016) as well as natural emissions and re-emissions inventory (4° × 5°) from Horowitz et al. (2017) (see Fig. 1).

2.      The multi-scale comparison presented requires additional evidence to support the conclusions and more information on data and assumptions. Fig. 3 must include results from all three scales. The article must describe the spatial and temporal surrogates and processes used to allocate emissions from each sector. The authors are strongly

encouraged to present maps of total and sectoral emissions maps at each scale in the supplemental materials. In Fig. 4, panels 2 and 3 appear identical—this should be addressed to provide evidence of different resolutions, spatial patterns, and the differences in magnitude described in the manuscript. The discussion (206-213) should reflect the roles of the emissions inventory and downscaling as limiting factors in resolution for this study rather than an inherent process resolution issue below 50 km.

- Multiscale of Hg surface concentration are shown in Fig. 3 now.

**Fig. 2** Comparison of monthly average Hg surface concentration of $Hg^0$ (top), $Hg^2$ (middle), HgP (bottom) from July to September 2013. The left panel is GEOS-Chem $4° \times 5°$ simulation. The other two columns from left to right correspond to different WFF-GC resolutions: 50 km $\times$ 50 km, 25 km $\times$ 25 km. Dots on the top row of the panel represents $Hg^0$ observation data from AMNet of NADP (http://nadp.slh.wisc.edu/AMNet/).

- For emission inventories, WRF-GC uses HEMCO (Lin et al, 2021). HEMCO is a software component designed for computing emissions. Users are allowed to select an ensemble of emission inventories, as well as state-dependent emission algorithms with the ability to re-grid, combine, over-write, scale, etc. emissions from various inventories by a configuration file, even adjusting model species with proper units. No further modification is needed to the model source code. Therefore, we do not need to do like using WRF-Chem, etc. to prepare emission files. HEMCO can do work, like linear

interpolation for simulations, automatically, so this is an advantage for GEOS-Chem and WRF-GC.

- Thanks for pointing this out. We also recognize this problem. Simply increasing resolution is only better for meteorology simulation because it can help the model to resolve small-scale weather conditions, but it is not that useful for emission. Since the resolution of emission inventories is fixed, the higher Hg wet deposition at higher resolution is mainly caused by meteorological precipitation but not emission.

Line 231-234: Here the increase of resolution is only better for meteorology simulation because the finer resolution can help the model resolve small-scale weather conditions. Since the resolution of emission inventories is fixed ($1° \times 1°$ and $4° \times 5°$), with higher resolution, more Hg wet deposition will be shown in our result because more convective precipitation is captured by the model.

3. The arbitrary filter of 70% data availability for site inclusion should be justified, and other observational data QA/QC steps described.
We chose this threshold following Holmes et al. (2010). The MDN and AMNet data have been quality checked before releasing. For example, all MDN data have a quality index like "A", "B", "C". ("D" is omitted).

Line 162: We only take sites that contain at least 75% availability of data for three months (Holmes et al., 2010).

4. The article and the GitHub and Zenodo repositories include all files necessary and sufficient for reasonable reproducibility. Addition of a WPS pre-processor file for the domains and scripts for the analysis and figures presented would bring the study close to full reproducibility.
Thanks for your advice. The whole WRF-GC model is too large to upload the whole folder. For parent model WRF-GC, Lin et al, 2020 and GitHub repository (https://doi.org/10.5281/zenodo.3550330) provided a detailed guide of porting WRF-GC model. Our research is based on the WRF-GC model. Anyone interested can first install WRF-GC model (https://fugroup.org/index.php/Installing_WRF-GC) and replace the necessary file from our GitHub repository. You may set your desired domain area and simulation period, while the other settings are the same as a normal WRF-GC simulation setup.

Line 325-326: The latest WRF-GC-Hg v1.0 is permanently archived at https://doi.org/10.5281/zenodo.6366777 (last access: 18th Mar 2022).

Reference:
Christopher Emery, Zhen Liu, Armistead G. Russell, M. Talat Odman, Greg Yarwood & Naresh Kumar (2017) Recommendations on statistics and benchmarks to assess photochemical model performance, Journal of the Air & Waste Management Association, 67:5, 582-598, DOI: 10.1080/10962247.2016.1265027

Lin, H., Feng, X., Fu, T.-M., Tian, H., Ma, Y., Zhang, L., Jacob, D. J., Yantosca, R. M., Sulprizio, M. P., Lundgren, E. W., Zhuang, J., Zhang, Q., Lu, X., Zhang, L., Shen, L., Guo, J., Eastham, S. D., and Keller, C. A.: WRF-GC (v1.0): online coupling of WRF (v3.9.1.1) and GEOS-Chem (v12.2.1) for regional atmospheric chemistry modeling – Part 1: Description of the one-way model, 13, 3241–3265, https://doi.org/10.5194/gmd-13-3241-2020, 2020.

Lin, H., Jacob, D. J., Lundgren, E. W., Sulprizio, M. P., Keller, C. A., Fritz, T. M., Eastham, S. D., Emmons, L. K., Campbell, P. C., Baker, B., Saylor, R. D., and Montuoro, R.: Harmonized Emissions Component (HEMCO) 3.0 as a versatile emissions component for atmospheric models: application in the GEOS-Chem, NASA GEOS, WRF-GC, CESM2, NOAA GEFS-Aerosol, and NOAA UFS models, 14, 5487–5506, https://doi.org/10.5194/gmd-14-5487-2021, 2021.

Holmes, C. D., Krishnamurthy, N. P., Caffrey, J. M., Landing, W. M., Edgerton, E. S., Knapp, K. R., and Nair, U. S.: Thunderstorms increase mercury wet deposition, 50, 9343–9350, https://doi.org/10.1021/acs.est.6b02586, 2016.

Bullock, O. R., and Brehme, K. A.: Atmospheric mercury simulation using the CMAQ model: formulation description and analysis of wet deposition results, Atmospheric Environment, 2135–2146 pp., 2002.

Gencarelli, C. N., de Simone, F., Hedgecock, I. M., Sprovieri, F., and Pirrone, N.: Development and application of a regional-scale atmospheric mercury model based on WRF/Chem: A Mediterranean area investigation, 21, 4095–4109, https://doi.org/10.1007/s11356-013-2162-3, 2014.

Pan, L., Lin, C. J., Carmichael, G. R., Streets, D. G., Tang, Y., Woo, J. H., Shetty, S. K., Chu, H. W., Ho, T. C., Friedli, H. R., and Feng, X.: Study of atmospheric mercury budget in East Asia using STEM-Hg modeling system, 408, 3277–3291, https://doi.org/10.1016/j.scitotenv.2010.04.039, 2010.

Holmes, C. D., Jacob, D. J., Corbitt, E. S., Mao, J., Yang, X., Talbot, R., and Slemr, F.: Global atmospheric model for mercury including oxidation by bromine atoms, 10, 12037–12057, https://doi.org/10.5194/acp-10-12037-2010, 2010.